# Radiotherapy Advances in Renal Disease—Focus on Renal Ischemic Preconditioning

**DOI:** 10.3390/bioengineering10010068

**Published:** 2023-01-05

**Authors:** Badr Khbouz, Shiyang Gu, Tiago Pinto Coelho, François Lallemand, François Jouret

**Affiliations:** 1Groupe Interdisciplinaire de Génoprotéomique Appliquée (GIGA), Cardiovascular Sciences, University of Liège (ULiège), 4000 Liège, Belgium; 2III. Department of Medicine, University Medical Center Hamburg-Eppendorf, 20251 Hamburg, Germany; 3Zhongshan Hospital, Fudan University, Shanghai 200032, China; 4Cyclotron Research Center, University of Liège, 4000 Liège, Belgium; 5Division of Radiotherapy, CHU of Liège, University of Liège (CHU ULiège), 4000 Liège, Belgium; 6Division of Nephrology, CHU of Liège, University of Liège (CHU ULiège), Avenue Hippocrate, 13, 4000 Liège, Belgium

**Keywords:** irradiation, radiotherapy, renal preconditioning, renal ischemia-reperfusion, acute kidney injury

## Abstract

Ionizing irradiation is widely applied as a fundamental therapeutic treatment in several diseases. Acute kidney injury (AKI) represents a global public health problem with major morbidity and mortality. Renal ischemia/reperfusion (I/R) is the main cause of AKI. I/R injury occurs when blood flow to the kidney is transiently interrupted and then restored. Such an ischemic insult significantly impairs renal function in the short and long terms. Renal ischemic preconditioning (IPC) corresponds to the maneuvers intended to prevent or attenuate the ischemic damage. In murine models, irradiation-induced preconditioning (IP) renders the renal parenchyma resistant to subsequent damage by activating defense pathways involved in oxidative stress, angiogenesis, and inflammation. Before envisioning translational applications in patients, safe irradiation modalities, including timing, dosage, and fractionation, need to be defined.

## 1. Introduction

Ionizing radiation is the energy released by atoms, which travels in the form of electromagnetic waves (gamma or X-rays) or particles (neutrons, beta, or alpha). It has been recognized as a risk factor for cancer in cellular and animal models, as well as in large epidemiological studies of populations exposed to extremely high doses of radiation [1,2,3]. Nevertheless, in recent decades, the use of ionizing irradiation as a therapeutic treatment has been widely applied. In particular, radiotherapy (RT) has become a fundamental tool against several diseases, especially some forms of cancer [4]. 

In the context of kidney diseases, several studies based only on animal models showed a protective effect of radiation therapy. In this review, we particularly focus on acute kidney injury (AKI), a known global public health problem that affects ~1000 people per million/year and is associated with a significant mortality rate [5,6]. Renal ischemia/reperfusion (I/R) injury is the leading cause of AKI [7] and corresponds to the transient interruption of the renal blood flow, with an abrupt drop in oxygen pressure and nutrient delivery leading to vascular and tubular dysfunction [8,9,10]. Here, we will provide an overarching view of recent advances in radiation therapy relevant to renal I/R and kidney diseases, as suggested in various murine models. We eventually envision the challenges and limitations of RT in the management of AKI in patients.

## 2. Renal Ischemia-Reperfusion

Renal I/R is an unavoidable event in kidney transplantation and cardio-thoracic surgery, with a negative impact on short- and long-term kidney outcomes [11,12]. Renal blood flow at rest is 1200 mL/min, which corresponds to 20% of cardiac output, while both kidneys represent less than 1% of total body weight [13]. Under physiological conditions, the renal blood flow remains constant up to a systolic arterial pressure of 80 mmHg due to the fine regulation of the sympathetic nervous system, the hormonal renin/angiotensin/aldosterone axis, and the synthesis of prostaglandins [13]. The prolonged interruption of renal blood perfusion leads to the cessation of nutrient supply and a drop in the partial pressure of oxygen. This transient interruption/reduction of renal blood flow, followed by its restoration and re-oxygenation, causes a cascade of cellular and tissue events grouped under the term “ischemic damage” [14,15]. 

Different I/R mouse models are used to mimic different aspects of the pathophysiology. Currently, two kinds of warm renal I/R models are mainly used: bilateral renal and unilateral renal I/R [16]. The unilateral renal I/R model has the advantage of a left nephrectomy being undertaken at the time of surgery. The left nephrectomy tissue serves as valuable control tissue in studies involving a pretreatment step, like irradiation, that modulates certain pathways [17,18,19]. The uniqueness of this model also makes it important for adhering to the three Rs rule (Replacement, Reduction, and Refinement) for a more ethical use of animals in research (Figure 1).

## 3. Renal Ischemic Preconditioning

Renal ischemic preconditioning (IPC) is an evolving concept that regroups all maneuvers aimed at preventing or attenuating the severity of the renal I/R injury. The principle of the “original” IPC was to mechanically expose the organ to brief episodes (3–5 min) of ischemia before conservation in prolonged ischemia in order to reduce ischemic damage and accelerate functional recovery [20,21]. In vivo, the renal IPC makes it possible to maintain the organization of the actin network of the cytoskeleton and the cell polarity, thereby preserving the polarized distribution of ion transporters such as the basolateral Na+/K+-ATPase. Tissue infiltrations by leukocytes, as well as the degree of apoptosis and necrosis, are significantly lower when IPC has been performed [22].

On the basis of the identified cellular targets of key pathways in renal IPC, the concept of pharmacological conditioning has emerged in the field of renal I/R. The use of pharmacological agents aims at stimulating the biochemical pathways of IPC [23]. Several drugs have been hypothesized to reproduce these protective mechanisms and have subsequently been tested in different settings (Table 1).

The current therapeutic strategies are directed to interact with the major I/R signaling pathways, such as inflammation, vascularization, energy metabolism, or oxygen transport [23,24,25].

## 4. Radiation-Induced Effects on the Kidney

The impacts of radiation on the tissue and/or organs depend on the dose of radiation received and absorbed. It is expressed in gray units (Gy). The potential impact of an absorbed dose depends on the type of radiation and the variable sensitivity of organs [26]. At the cellular level, the effects of ionizing radiation can manifest as cell death or changes in the cytogenetic information [26]. These events can lead to adverse tissue reactions, in which manifestations depend on exceeding the dose threshold, or to stochastic effects, when the effect increases with the dose (Figure 2).

Basically, ionizing radiation can cause double-stranded breaks (DSB) in the DNA, followed by cell death including apoptosis and necrosis of renal endothelial, tubular, and glomerular cells in the case of radiation nephropathy (RN) [27]. Kidney toxicity depends on the use and intensity of RT. Protective genetic and biochemical effects can also be induced by DNA damage response (DDR) activation after sensing the DSB. This system reacts with cell death induction or cell cycle arrest and DNA repair based on the affected tissue and the severity of the damage. The oxidative stress [28], the renin–angiotensin (RAS) system [29], cellular senescence [30], proliferation, angiogenesis [18,31,32], and endothelial and hemodynamic modulations [33] might be involved in the putative mechanisms in radiation-induced effects on the renal parenchyma. 

## 5. Radiation-Based Therapy in Kidney Diseases

Radiation therapy relies on ionizing radiation and is usually used as part of cancer treatment to control or eradicate malignant cells [34]. Irradiation-mediated renal IPC, although not yet well reported, would aim to render cells or tissues resistant to subsequent damage by activating their intracellular defense system [35]. Indeed, despite the potential for radiation exposure to cause kidney damage, a certain dose of irradiation has also been shown to have a protective effect in animal studies. A summary of the literature suggesting the renal protective role of kidney-centered ionizing irradiation in rodent models of kidney diseases is listed in Table 2. Note that there is no translational data in humans.

In the global context of kidney diseases, it has been shown that continuous whole-body low-dose-rate gamma irradiation ameliorates diabetic nephropathy and increases the lifespan in mice through the activation of renal antioxidants [41]. Taylor et al. also showed that the incidence of kidney disease was significantly reduced following a 10 mGy irradiation compared to non-irradiated mice [42]. Histopathological changes in the renal parenchyma of several animal models of kidney diseases showed a protective effect after local or whole-body irradiation to the kidneys, such as in CKD [37], crescentic nephritis [38], and diabetic nephropathy rodent models [40]. Aunapuu et al. found that the CKD rat model through 5/6 nephrectomy (NPX) manifested a high level of proliferating and apoptotic markers, which could be reduced significantly in glomeruli and distal tubular cells following a 3 Gy dose of γ radiation on the left kidney [36]. The indicators of renal function, such as proteinuria, systolic blood pressure, and serum creatinine (SCr), were also improved after the irradiation in these kinds of CKD rodent models [36,37]. Anti-oxidative stress, anti-inflammatory, and anti-renal fibrosis may be considered as the potential benefits of irradiation therapy. 

Cheng et al. [40] found that the whole-body exposure of type 1 diabetic mice (DM) to 25 mGy X-rays weekly decreased the microalbuminuria, the renal accumulation of 3-nitrotyrosine and 4-hydroxynonenal, and the renal expression of collagen IV and fibronectin, suggesting that low-dose radiation (LDR) improved DM-induced oxidative/nitrosative damage and renal fibrosis effectively. Shao et al. [43] also showed that frequent LDR X-ray treatment attenuated dyslipidemia and insulin resistance followed by renal inflammation and oxidative stress in mice with type 2 diabetes. A low dose of gamma radiation has also been found to protect against D-GalN-induced renal damage in Swiss albino rats. The results revealed increases in antioxidant activities and decreases in inflammatory markers (tumor necrosis factor-alpha (TNF-α) and nuclear factor kappa-light-chain-enhancer of activated B cells (NF-κB). Additionally, this treatment was associated with an upregulation of *Nrf-2* gene expression, which relates to the prevention of oxidative stress, and decreased lipid peroxidation levels, which were concordant with histopathological findings following the irradiation therapy [39]. In an experimental model of crescentic nephritis in rats [38], the overexpression of active caspases 3 and 7, elevation in the TUNEL assay, and decrease in PCNA were noted after local, bilateral irradiation treatment. The improvement in morphology, the decrease in the proliferation marker, and the elevation in the apoptosis assay have also been shown in a CKD rat model [37]. 

In the context of I/R diseases, several studies have reported a protective effect of radiation therapy against I/R injury. Yuan-Po et al. showed that far-infrared radiation attenuated I/R injury in rat testis by inducing heme oxygenase (HO1) expression [44]. Lakyova et al. similarly demonstrated that low-level laser irradiation causes a profound reduction in the amount of necrotic tissue and enhances recovery after I/R muscle injury in rat hindlimbs by attenuating the inflammatory reaction and facilitating angiogenesis [45]. 

## 6. Irradiation Preconditioning against AKI

Interestingly enough, in the context of in vivo studies of ischemic AKI using renal I/R mouse models, whole-body irradiation (WBI) has been suggested to induce renal IPC. Indeed, the 8 Gy WBI significantly attenuated the elevations of SCr and BUN concentrations, structural damage, lipid peroxidation, expression, and activity of NADPH oxidase (NOX)-2, nitrotyrosine levels, and hydrogen peroxide production following renal I/R [35]. This protection was inhibited by the treatment of animals with manganese (III) tetrakis (1-methyl-4-pyridyl) porphyrin (MnTMPyP), a superoxide scavenger, indicating that irradiation-induced preconditioning (IP)-rendered protection was triggered by superoxide formation, enhancing manganese superoxide dismutase (MnSOD) activity and expression, and HSP-27 expression. 

Advances in conformational radiology and preclinical radiotherapy research have recently spurred the development of precise micro-irradiators for small animals, including rodents. These devices are often kilovolt X-ray radiation sources combined with high-resolution computed tomography (CT) imaging equipment for image guidance, as the latter allows precise and accurate beam positioning. These devices are similar to modern human radiotherapy machines and are considered a major step forward in radiobiology research [46,47]. 

In order to investigate the impact of kidney-focused irradiation in mice before renal I/R, we have recently used the small animal irradiator and scanner (SmART) instrument from precision X-ray (North Branford, CT) designed to image, target, and irradiate cells and small animals up to the size of rodents [48,49,50]. This scanner provides images that have a resolution of 0.1 mm. A prescan of the whole body helped to precisely locate the volume to be investigated, and this volume was then defined in all three dimensions by moving cursors on the computer screen with the mouse. The radiation exposure focused on the kidneys, with beams of 225 KV and 13 mA (Figure 3A,B). 

## 7. Kidney-Centered Irradiation Mediates IPC

The bilateral X-ray renal irradiation with a single dose of 8 Gy induced IPC in mice, with reduced macrophage infiltration and attenuated tubular necrosis following renal I/R. The metabolomics signature of renal I/R was attenuated in pre-irradiated mice. BUN and SCr plasma assays showed stable circulating levels of these two AKI biomarkers, which reflect preserved renal function in pre-irradiated mice exposed to I/R. Such a renal IPC was significantly observed as early as 14 days post-irradiation, preceding the I/R episode. Moreover, the physiological follow-up study of irradiated mice showed no structural or functional complications 3 months after kidney-centered irradiation. The comparative high-throughput RNA-seq between the irradiated and control kidneys reveals that renal irradiation was associated with an upregulation of signaling pathways involved in angiogenesis, cell proliferation, and stress response and a downregulation of the JNK cascade [18].

Several works reported a protective effect of radiation therapy against I/R injuries [36,37]. By extrapolating from these studies, one may speculate that even the exposure of animals to low doses (0.1–2 Gy) could have a similar degree of protective effect in the case of renal I/R. Recently, multiple exposures to low-dose whole body irradiation (LD-WBI) in C57BL/6J mice have been reported to significantly suppress the diabetes-induced systemic, renal inflammatory response, and renal oxidative damage, thereby preventing renal dysfunction and fibrosis [40]. Aunapuu et al. also showed that low-dose-rate gamma irradiation slows the process of CKD through the reduction of apoptotic cells [41]. With the application of higher irradiation doses like the one used in our study, acute toxic effects and potentially late long-term effects could be induced. In the perspective of therapeutic irradiation in humans, future studies will be required to determine whether kidney-centered low-dose irradiation at decreasing doses (upon a 10-based order of magnitude) could have a similar protective impact against renal I/R injury. In our murine model, the post-irradiation follow-up at 3 months did not reveal complications regarding the physiological functions of the irradiated mice or the establishment of renal fibrosis. However, this follow-up study did not go beyond 3 months after irradiation. A larger study, including the evaluation of overall physiological parameters with a longer follow-up period, is therefore required. 

Another important issue in radiotherapy concerns the therapeutic window. In the work of Kim et al., WBI with 8 Gy induced IPC against renal I/R 6 days *after* irradiation [35]. In our kinetics study of pre-irradiation conditioning, we compared the impact of kidney-focused irradiation 7, 14, and 28 days *before* renal I/R. The IPC was significantly reached at day 14, and a trend was observed at day 7 [18]. As a result, future research will concentrate on characterizing the kinetics of irradiation-mediated IPC from 7 days to 1 day after renal irradiation in order to mimic clinical settings as closely as possible, such as kidney transplantation or cardiothoracic surgery.

## 8. Conclusions

AKI is a growing medical challenge with a high incidence that affects numerous patients worldwide. The occurrence of AKI is associated with an increased complication rate and the development of CKD, as well as short- and long-term mortality. The reported demonstrations highlight new evidence for the development of innovative therapeutic approaches. Still, we need to further improve the safety of the efficacy/toxicity ratio. Taken together, the modulation of the molecular targets of irradiation-induced renal IPC may open new avenues in the development of innovative pharmacological strategies against a major medical problem, which is acute ischemic kidney injury.

## Figures and Tables

**Figure 1 bioengineering-10-00068-f001:**
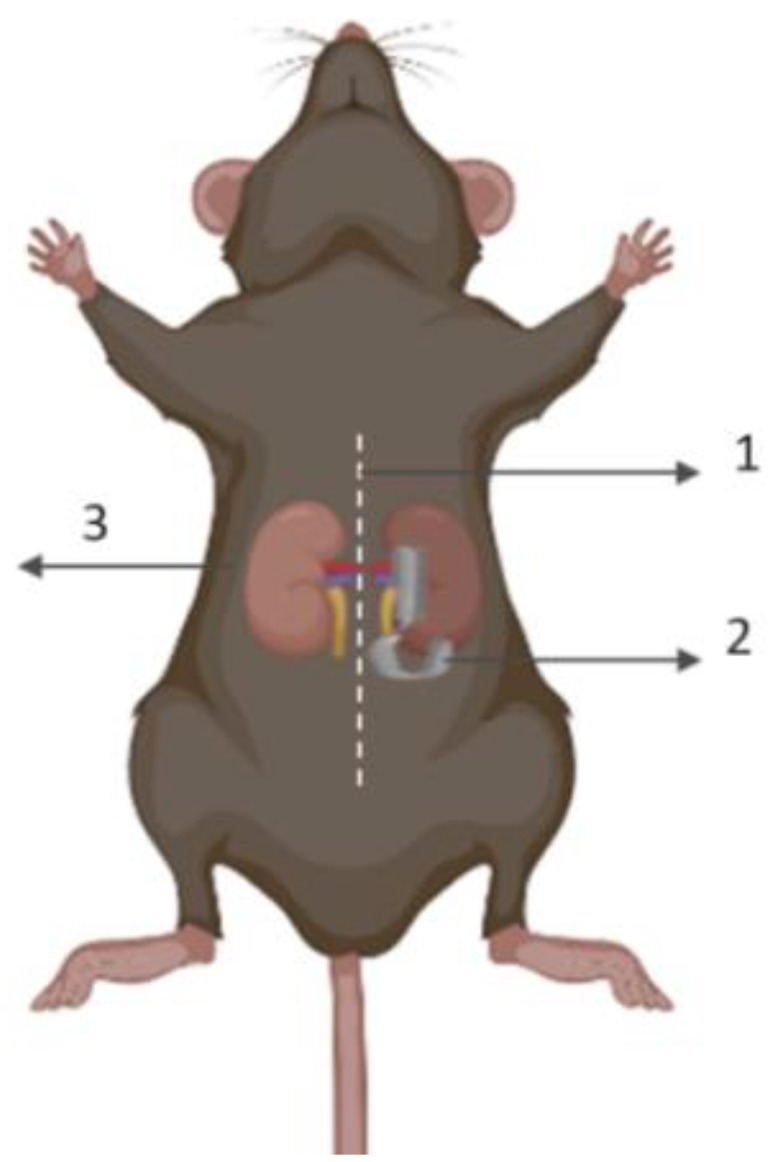
Mouse model of I/R-induced AKI. (1) Laparotomy, followed by (2) 30 min of right renal ischemia, in parallel to (3) left nephrectomy (control sham). Forty-eight hours after reperfusion, blood and kidneys are harvested for the assessment of the ischemic damage. (Created with BioRender.com).

**Figure 2 bioengineering-10-00068-f002:**
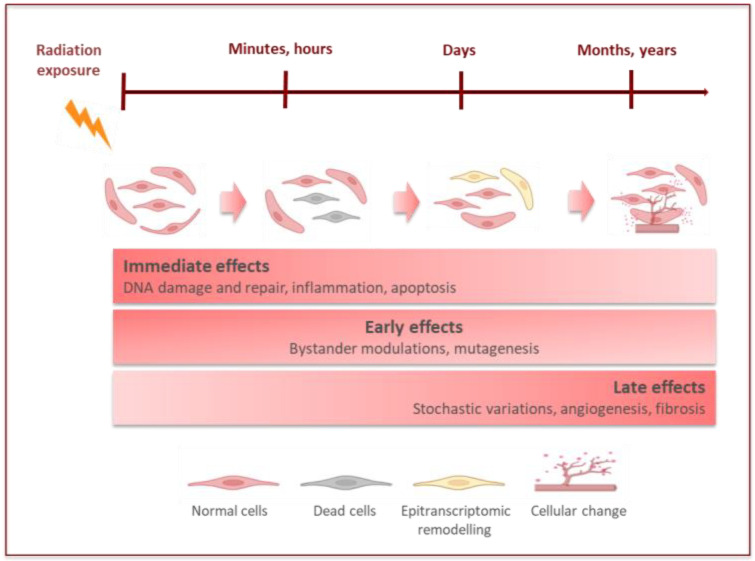
Onset of radiation-induced effects at cellular and tissue levels. Immediate radiation effects lead to signaling events such as DNA damage and repair mechanisms, tissue inflammation, and apoptosis (left). Moreover, radiation is associated with bystander effects (biological effects expressed by non-irradiated cells through signals produced by the irradiated area) and (epi)genomic modulations (middle). The cellular epigenetic modifications can trigger particular circuits after radiation exposure (right). (Created with BioRender.com).

**Figure 3 bioengineering-10-00068-f003:**
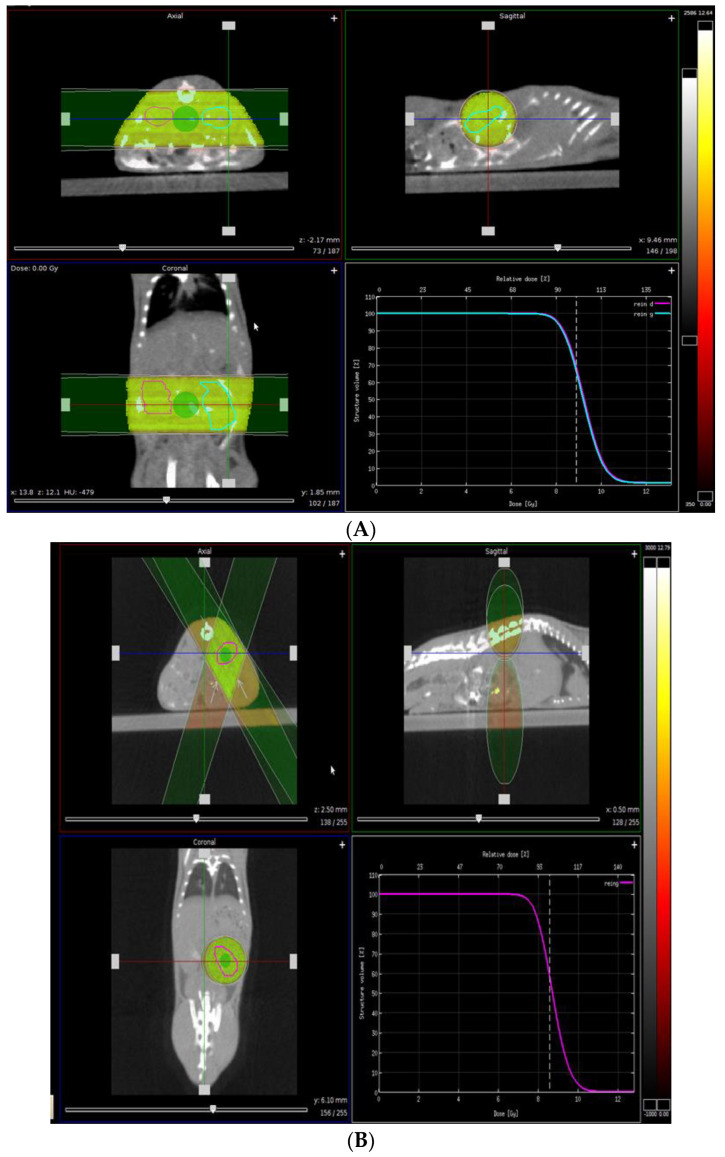
PXi X-Rad SmART renal radiation therapy: (**A**) bilateral renal irradiation with two beams of X-rays targeting both kidneys and (**B**) unilateral renal irradiation with three beams of X-rays targeting the left kidney.

**Table 1 bioengineering-10-00068-t001:** Examples of pharmacological renal ischemic preconditioning in rodents. MAPK: mitogen-activated protein kinases; GPCR: G protein-coupled receptor; HIF-1: hypoxia-inducible factor 1; NOS: nitric oxide synthase; and AMPK: AMP-activated protein kinase (adapted from [23]).

Drugs	Protein Targets	Pathways
Cyclosporine; FK-506 Mesenchymal stem cells (MSC); Adenosine; Apyrase; and Catecholamines	MAPK GPCR	Inflammation
Erythropoıetin; Isoflurane	HIF-1a	Hypoxia
L-carnitine; Lithium; Danshen; N-acetylcysteine; and Spermine NONOate	Free radicals; NOS	Oxidative stress
Metformin; AICAR; and Hemin	AMPK Hmox1 inducer	Metabolism

**Table 2 bioengineering-10-00068-t002:** Summary of the literature exhibiting renal protection following ionizing irradiation on different assays at different doses in rodent experimental disease models: monocyte chemoattractant protein-1(MCP-1), dUTP nick-end labeling (TUNEL), nuclear factor erythroid 2-related factor 2 (Nrf-2), urinary microalbumin (Malb), blood urea nitrogen (BUN), serum creatinine (SCr), 3-nitrotyrosine (3-NT), 4-hydroxynonenal (4-HNE), superoxide dismutase (CuZnSOD), manganese superoxide dismutase (MnSOD), glutathione peroxidase GPx, heme oxygenase 1 (HO1), heat shock protein (HSP), platelet and endothelial cell adhesion molecule 1 (PECAM1), collagen IV (Col IV), ↑: significant functional and morphological improvement, and ↓: significant decrease in degradation markers.

Disease Model	Radiation	Site	Dose	Time	Aspect	Tendency	Reference
Adult Wistar rats (CKD)	Γ	Left kidney	3 Gy	2 w	Function (urinary proteins excretion rate, awake systolic blood pressure, and SCr) Morphology MCP-1	↑ ↑ ↓	[36]
Adult Wistar rats (CKD)	Γ	Left kidney	3 Gy	1/2 w	Function (Urinary proteins, systolic blood pressure, and SCr) Morphology Apoptosis (TUNEL)	↑ ↑ ↓	[37]
Inbred male Sprague- Dawley rats (Crescentic nephritis)	X	Local bilateral kidneys	0.5 Gy	1/2/3/4 w	Function (SCr) Morphology PCNA ED-1 Apoptosis (Caspase 3/7 and TUNEL)	↑ ↑ ↓ ↑	[38]
Adult male C57BL/6 mice (I/R induced AKI)	X	Whole body	8 Gy	1 w	Function (BUN and SCr) Morphology Oxidative stress (MnSOD and HSP27)	↑ ↑ ↑	[35]
Adult male Swiss albino rats (D-GalN-induced renal damage)	Γ	Whole body	0.25 Gy	N/A	Antioxidant activities (CuZnSOD and GPx) Lipid peroxidation level Function Inflammation (TNF-a and NF-KB) Morphology Nrf-2 gene	↑ ↓ ↑ ↓ ↑ ↓	[39]
Adult male C57BL/6J mice (Type I diabetes)	X	Whole body	12.5 or 25 mGy	4/8 w	Function (SCr and urinary microalbumin) Morphology (PAS staining) Nitrosative damage (3-NT and 4-HNE) Renal fibrosis (Col IV and fibronectin)	↑ ↑ ↓ ↓	[40]
Adult male C57BL/6 mice (I/R-induced AKI)	X	Local bilateral kidneys	8 Gy	2 w	Function (BUN and SCr) Morphology (PAS staining) Inflammation (Cd11b; F4/80) Angiogenesis (PECAM1) Oxidative stress (HSP70; HO1)	↑ ↑ ↓ ↑ ↑	[18]

## Data Availability

Data sharing is not applicable.

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
