# Peer review of "Radiotherapy Advances in Renal Disease—Focus on Renal Ischemic Preconditioning"

_bioengineering, 2023, doi:10.3390/bioengineering10010068_

Round 1

Reviewer 1 Report

This is generally an interesting review about the possibilities of radiotherapy in kidney disease based on animal studies. And that’s what it is, it looks at some aspects based on animal studies. This is not a bad thing or lessens the value, in my opinion the paper should just be structured accordingly. The authors tried to include a lot of aspects from basic physics to nuclear bombs to terms used in radiation protection to clinical radiotherapy that has nothing to do with these models. Until I reached chapter 5 (btw check the numbering) I was a little annoyed as it was jumping from aspect to aspects explaining supposedly random (later it made sense) aspects in more or less detail without the reader understanding why. I would generally recommend to re-structure the manuscript to make it more on point and concise. There is limited point in my opinion to widely speculate about clinical aspects when we are talking about preclinical animal models with a short follow-up. And I would really restructure the introduction so that readers understand clear and quickly the purpose. 

Some detailed aspekts:

Introduction

Line 30 – 36: Although this basic pyhsics and and wordwide distribution of radioactive material part is correct, I am not sure what that has to do with radiotherapy induced kidney damage

General: See general comments, we are talking about preclinical animal data. That has to be clear for the reader.

I would wish for the introduction to be more related to the title of purpose of the paper and guide the reader to the topic at hand

CHapter 2 

See general comments. For me this should have a concise line rather than jumping from aspekt to aspect that have something to do with Radiotherapy or Kidney injury.

Chapter 3 

See general comments. For me this should have a concise line rather than jumping from aspekt to aspect that have something to do with Radiotherapy or Kidney injury.

Chapter 4

line 104: The sentence about Sv is correct, but one uses usually Gy in radiation therapy and Sv in radiation protection. Therefore I don’t see a benefit of adding this sentence 

line 116: it can and it does cause double strand breaks, not could

General could use restructuring as to molecular, cellular, microscopic, macroscopic changes, Be more concrete about effects on kidney. Most of that is basic mechanisms in most cells/organs, again, see the general comment about having a clear line

Chapter 5

lines 127 – 130: Yes, but what has that to do with the kidney?

Line 134 – protective animal studies with a wide dose range 

Chapter 7

lines 201 – 215: What has that to do with AKI? Same with figure 3. This is about the experimental setup and technical possibilities 

Conclusion

Again, we are talking about preclinical animal data with rather short FU compared to the lifespan of humans. “The treatment of kidney diseases with RT continues to evolve.” is maybe a little too strong given the evidence at hand. 

Reviewer 2 Report

A known global public health problem is related to Acute Kidney Injury (AKI) with major morbidity or even mortality. The main cause of AKI is renal Ischemia/Reperfusion (I/R). I/R injury occurs when blood flow to the kidney is transiently interrupted and then restored. Such an ischemic insult significantly harms renal function in the short term and long term. The paper discusses the 3-8 Gy gamma radiation dose effects on AKI. 

Reviewer 3 Report

I read with great interest the Review Article by Badr Khbouz et al. entitled “Radiotherapy Advances in Renal Disease - Focus on Renal Ischemic Preconditioning”. The manuscript is well written, interesting and timely.

However, the following minor observations are made:

·      As the Authors stated “Ionizing irradiation is widely applied as a fundamental therapeutic treatment in several diseases.”, they should briefly discuss its potential use, benefits and drawbacks and then focus their review to acute kidney injury.

·      The Authors should report how can be translated findings in “murine models” to “humans”. Otherwise, they should better specify that their Review Article is aimed at only describing pathological states in murines.

·      Minor orthographical and grammatical errors have been found throughout the manuscript.
